# Anticancer Activity of (*S*)-5-Chloro-3-((3,5-dimethylphenyl)sulfonyl)-*N*-(1-oxo-1-((pyridin-4-ylmethyl)amino)propan-2-yl)-1*H*-indole-2-carboxamide (RS4690), a New Dishevelled 1 Inhibitor

**DOI:** 10.3390/cancers14051358

**Published:** 2022-03-07

**Authors:** Antonio Coluccia, Marianna Bufano, Giuseppe La Regina, Michela Puxeddu, Angelo Toto, Alessio Paone, Amani Bouzidi, Giorgia Musto, Nadia Badolati, Viviana Orlando, Stefano Biagioni, Domiziana Masci, Chiara Cantatore, Roberto Cirilli, Francesca Cutruzzolà, Stefano Gianni, Mariano Stornaiuolo, Romano Silvestri

**Affiliations:** 1Laboratory Affiliated with the Institute Pasteur Italy—Cenci Bolognetti Foundation, Department of Drug Chemistry and Technologies, Sapienza University of Rome, Piazzale Aldo Moro 5, 00185 Rome, Italy; antonio.coluccia@uniroma1.it (A.C.); marianna.bufano@uniroma1.it (M.B.); giuseppe.laregina@uniroma1.it (G.L.R.); michela.puxeddu@uniroma1.it (M.P.); 2Laboratory Affiliated with the Institute Pasteur Italy—Cenci Bolognetti Foundation, Biochemical Sciences “Rossi Fanelli”, Institute of Biology and Molecular Pathology of CNR, Sapienza Università di Roma, Piazzale Aldo Moro 5, 00185 Rome, Italy; angelo.toto@uniroma1.it (A.T.); alessio.paone@uniroma1.it (A.P.); amani.bouzidi@uniroma1.it (A.B.); francesca.cutruzzola@uniroma1.it (F.C.); stefano.gianni@uniroma1.it (S.G.); 3Department of Pharmacy, University of Naples “Federico II”, Via Domenico Montesano, 80131 Naples, Italy; giorgia.musto@unina.it (G.M.); nadia.badolati@unina.it (N.B.); mariano.stornaiuolo@unina.it (M.S.); 4Department of Biology and Biotechnologies “Charles Darwin”, Sapienza University of Rome, Piazzale Aldo Moro 5, 00185 Roma, Italy; viviana.orlando@uniroma1.it (V.O.); stefano.biagioni@uniroma1.it (S.B.); 5Department of Basic Biotechnological Sciences, Intensivological and Perioperative Clinics, Catholic University of Sacred Heart, Largo Francesco Vito 1, 00168 Rome, Italy; domiziana.masci@unicatt.it; 6National Center for the Control and Evaluation of Drugs, Istituto Superiore di Sanità, Rome, Viale Regina Elena 299, 00161 Rome, Italy; cantatore.1690075@studenti.uniroma1.it (C.C.); roberto.cirilli@iss.it (R.C.)

**Keywords:** cancer, DVL1, PDZ domain, WNT pathway, virtual screening

## Abstract

**Simple Summary:**

The WNT/β-catenin pathway regulates a huge number of cellular functions, and its dysregulation is correlated to the development of cancer. In this work, we focused on the interaction between Dishevelled 1 (DVL1) protein, an important player in this pathway, and its cognate receptor Frizzled via a shared PDZ domain. Computational studies led to the discovery of racemate RS4690 (**1**) showing selective inhibition of DVL1 binding. After separation of the racemic mixture, enantiomer (*S*)-**1** inhibited DVL1 with an EC_50_ of 0.49 ± 0.11 μM and the growth of HCT116 cells that did not present the APC mutation with an EC_50_ value 7.1 ± 0.6 μM, and caused a high level of ROS production. Compound (*S*)-**1** shows potential as a new therapeutic agent against WNT-dependent colon cancer.

**Abstract:**

Wingless/integrase-11 (WNT)/β-catenin pathway is a crucial upstream regulator of a huge array of cellular functions. Its dysregulation is correlated to neoplastic cellular transition and cancer proliferation. Members of the Dishevelled (DVL) family of proteins play an important role in the transduction of WNT signaling by contacting its cognate receptor, Frizzled, via a shared PDZ domain. Thus, negative modulators of DVL1 are able to impair the binding to Frizzled receptors, turning off the aberrant activation of the WNT pathway and leading to anti-cancer activity. Through structure-based virtual screening studies, we identified racemic compound RS4690 (**1**), which showed a promising selective DVL1 binding inhibition with an EC_50_ of 0.74 ± 0.08 μM. Molecular dynamic simulations suggested a different binding mode for the enantiomers. In the in vitro assays, enantiomer (*S*)-**1** showed better inhibition of DVL1 with an EC_50_ of 0.49 ± 0.11 μM compared to the (*R*)-enantiomer. Compound (*S*)-**1** inhibited the growth of HCT116 cells expressing wild-type APC with an EC_50_ of 7.1 ± 0.6 μM and caused a high level of ROS production. These results highlight (*S*)-**1** as a lead compound for the development of new therapeutic agents against WNT-dependent colon cancer.

## 1. Introduction

The Wingless/integrase-1 (WNT) β-catenin pathway controls organogenesis and patterning in developing embryos, while allowing somatic stem cell maintenance and differentiation in adult organisms [1]. In humans, dysregulation of WNT β-catenin signaling leads to cancer cell proliferation [2], de-differentiation, and cancer stem cell genesis [3]. By virtue of its importance in physiological conditions, anticancer drugs acting as specific negative modulators of the pathway should be preferred to pure inhibitors of the WNT-β-catenin signaling. However, despite its involvement in a high number of malignancies, the identification of drugs finely tuning the WNT pathway seems hard to achieve. The difficulty in finding these modulators arises from the multitude of interconnected signaling branches and proteins involved in the WNT signaling cascade [4].

In the “canonical” WNT β-catenin pathway, the binding of morphogen WNTs to Frizzled Receptors (FZDs) recruits the PDZ family member Dishevelled (DVL) at the plasma membrane (PM) [5]. DVL is a 736 amino acid long scaffold protein made of three consecutive domains, namely DIX, PDZ and DEP. The DIX domain is constituted by 80–85 amino acids and contributes to DVL oligomerization. The 80–90 amino acid long PDZ domain binds to: (i) a PDZ binding motif (a conserved KTXXXW signaling sequence) allocated at the C-terminal tail of the receptors; (ii) a discontinuous domain spanning through the intracellular loops of the receptors. Finally, the 90–100 amino acid long DEP domain binds to the C-terminal tail of FZD and to its intracellular loops, and it is involved in the intracellular transduction of the WNT β-catenin signaling [6]. The WNT–FZD–DVL complex inactivates the destruction complex formed by Glucose Synthase Kinase 3β (GSK-3β), Axin, and Adenomatous Polyposis Coli (APC) [7] and leads to the intracellular accumulation and nuclear translocation of β-catenin [8].

In the nucleus, β-catenin binds the protein T-cell factor 1 (TCF-1) and regulates TCF/lymphoid enhancer binding factor (LEF)-dependent transcription of WNT/β-catenin target genes such as *cyclin D1*, *c-myc*, and *lgr5*, resulting in eliciting the development and progression of some solid tumors and hematological malignancies. Thus, DVL acts by switching on the WNT, a signals cascade that, in pathological conditions, results in transcription of oncogenes. According to the evidence, it was also reported that RNA interference and knockdown of DVL arrest the tumoral growth [9,10].

DVL proteins comprise three orthologues, DVL-1, DVL-2 and DVL-3, that are highly conserved to each other in sequence and folding. The DVL-2 isoform is the most abundant (95% of the total pool) and changes in its expression results with minimal effect on WNT signaling. On the other hand, canonical signaling was most sensitive to changes in the abundance of either DVL-1 or DVL-3 [11]. Despite the fact that DVL-1 is one of the least abundant of the total DVL protein pool, it has been shown to play an imperative role in stimulating phosphorylation and activation of the WNT transcriptional activity of the LDL receptor-related protein 6 (LRP6), an ortholog member of the LRP family [12,13]. The DVL-1 and DVL-3 proteins also play an important role in cancer chemoresistance. Both DVL-1 and DVL-3 are overexpressed in multidrug-resistant colorectal cancer cells compared to their parental cells. Silencing either DVL-1 or DVL-3 re-sensitizes resistant cells to multiple drugs including vincristine, 5-fluorouracil and oxaliplatin. [14,15] Moreover, DVL-1 and DVL-3 increase the levels of multidrug resistance protein 1 (P-gp/MDR1), multidrug resistance-associated protein 2 (MRP2), breast cancer resistance protein (BCRP), Survivin and Bcl-2, which are correlated with multidrug resistance. The interaction of the Frizzled receptor with DVL by the PDZ protein domain is crucial to allow the WNT signaling propagation [16].

Agents targeting specifically the PDZ protein domain of DVL have the potential to downregulate the WNT pathway, leading to inhibition of tumor growth. Recently, some peptides/peptidomimetics, small molecules and antibodies have been reported as DVL PDZ inhibitors [16,17]. FJ9 was the first non-peptide antagonist of DVL protein–protein interaction reported to suppress β-catenin–dependent tumor cell growth [18]. Representative examples of small molecules include sulindac [19], CalBioChem-322338 [20], N-benzoyl-2-aminobenzoic acid [21], phenoxyacetic acid [22] and indole-3-carboxamide [23] derivatives. Intensive effort is under way to develop selective small molecule modulators of the DVL PDZ domain as potential anticancer agents [4].

Taking into account the key role played by DVL1 in tumors and chemoresistance, we focused on the identification of new small molecules able to impair the DVL1 binding to its cognate receptor, FZD. Our group is widely involved in the identification of modulators of the WNT pathway. Recently, we reported FzM1 [24], a negative modulator of FZD4 (an FZD family member) by binding an allosteric site located in the intracellular loop 3 (ICL3), ultimately inhibiting DVL binding to FZD4. We have also reported FzM1.8 [25], a positive allosteric modulator of the WNT pathway, proving that fine tuning of the FzM1 scaffold results in dramatically different effects on the WNT pathway. Additionally, we recently identified 3-benzyl-5-chloro-*N*-(4-(hydroxymethyl)phenyl)-1*H*-indole-2-carboxamide (2) as an inhibitor of Na^+^/H^+^ exchanger 3 regulating factor 1 (NHERF1) by targeting the PDZ1 domain [26]. NHERF1 is well able to bind to FZDs; the interactions involve the same C-terminal region used by FZDs to form contacts with DVL through the PDZ domains. To our knowledge, DVL1 inhibitors with high selectivity toward NHERF1 have not been reported so far. We carried out a research project aimed at discovering selective DVL1 inhibitors. Since the knock-out of NHERF1 was reported to favor the DVL binding to FZDs [27], a selective DVL1 inhibitor would exhibit improved activity. In contrast, inhibition of both NHERF1 and DVL may result in weaker DVL inhibition. In this work, we performed structure-based virtual screening studies aimed at the discovery of DVL1 selective toward NHERF1 inhibitors. These studies led to the successful identification of compound RS4690 (**1**) (Figure 1) as a potent and selective inhibitor of DVL1. The pure enantiomer (*S*)-**1** showed greater inhibition of DVL1 than the (*R*)-enantiomer. Binding competition assays and biological evaluation in cells confirmed the mechanism of action, the block of the WNT pathway, and the anticancer activity.

## 2. Materials and Methods

### 2.1. Computational Studies

All molecular modelling studies were performed on a MacPro dual 2.66 GHz Xeon running Ubuntu 14 LTS. The PDZ structure of DVL1 was obtained by homology modelling. The template structures, PDZ of DVL2, were available at the PDB (pdb code 3CBZ, 3CBY and 3CC0) [28], while the PDZ of DVL1 sequence was retrieved from UniProt database (Available online: https://www.uniprot.org/ accessed on 9 February 2022) under the code O14640. The sequence alignment labels showed 75% sequence identity and 85% sequence similarity (positives). The hydrogen atoms were added to the protein using Maestro protein preparation wizard [29] and minimized, keeping all the heavy atoms fixed until an rmsd gradient of 0.05 kcal/(mol·Å) was reached. Ligand structures were built with Maestro and minimized using the MMFF94x force field until an rmsd gradient of 0.05 kcal/(mol·Å) was reached. The docking simulations were performed using PLANTS [30]. We set a binding lattice of 12 Å radius using all default settings. The pharmacophore model was obtained by Phase [31]. The pharmacophore features were fixed at the interaction points identified by compound **1** binding mode analyses. A hydrophobic query was also added in a sub-pocket lined by Leu12, Val75 and Ile81. The fitting to this query addressed the desired selectivity between DVL1 and NHERF1 PDZ domain. The polar features had a tolerance of 2 Å while the hydrophobic features had a tolerance of 2.5 Å. The training set obtained by docking computations was scored by the fitting to the pharmacophore model; the selected compounds must match at least five features, and hydrophobic and aromatic queries were equivalent. Molecular dynamics was performed with the AMBER 20 suite [32]. The inhibitors were parametrized by Leap of Amber. The structures of complexes were solvated in a periodic octahedron simulation box using TIP3P water molecules, providing a minimum of 10 Å of water between the protein surface and any periodic box edge. Ions were added to neutralize the charge of the total system. The water molecules and ions were energy-minimized, keeping the coordinates of the protein–ligand complex fixed (1000 cycles), and then the whole system was minimized (10,000 cycles). Following minimization, the entire system was heated to 298 K (20 ps). The production (10 ns) simulation was conducted at 298 K with constant pressure and periodic boundary conditions. Shake bond length condition was used (ntc = 2). Production was carried out on a Cineca Marconi 100 supercomputer. Trajectory analysis was carried out by CPPTRAJ program [33]. The ADME evaluations were carried out by swissADME server, a free web tool to evaluate pharmacokinetics, drug-likeness and medicinal chemistry friendliness of small molecules (36). The pictures reported in the manuscript were generated with PyMOL [34].

### 2.2. Cell Cultures

HEK293, HCT116, SW480, and SW620 cells were grown in DMEM (41965-039, GIBCO, Thermo Fisher Scientific, Waltham, MA, USA) supplemented with 10% FBS (10270, GIBCO, Thermo Fisher Scientific, Waltham, MA, USA), Glutamax (35050-061, GIBCO, Thermo Fisher Scientific, Waltham, MA, USA), and Pen/Strep (15070-063, GIBCO, Thermo Fisher Scientific, Waltham, MA, USA). HEK293 transfection was performed using Lipofectamine 2000 (11668019, Thermo Fisher Scientific, Waltham, MA, USA) following the manufacturer’s instructions. Mother stock solutions of compounds were obtained by dissolving them in DMSO at the dilution of 30 mM and storing at −80 °C.

### 2.3. Cell Viability

Cell viability was evaluated by XTT [35] (SW480 and SW620 cells) or MTT [36] (HEK293 and HCT116 cells) colorimetric assays. Briefly, cells (range 10–30 × 10^3^ cells/well) were seeded in 96-well microculture plates and then exposed to increasing concentrations of different compounds (range 0–300 µM) for 48 or 72 h. At the end of the treatment, media were removed and incubated at 37 °C in the dark for 4 h in phosphate-buffered saline (PBS) containing 0.2 mg/mL of sodium 3′-[1-[(phenylamino)-carbonyl]-3,4-tetrazolium]-bis(4-methoxy-6-nitro)benzene-sulfonic acid hydrate (XTT) (Thermo Fisher Scientific, Waltham, MA, USA) or (3-[4,5-dimethylthiazol-2-yl]-2,5 diphenyl tetrazolium bromide (MTT) (Merck, Kenilworth, NJ, USA) and phenazine methosulfate (PMS) at a final concentration of 25 µM. Absorbance at 450 nm and a reference wavelength of 650 nm was then measured using a microplate spectrophotometer (Multiskan FC Microplate Photometer, Thermo Scientific, Waltham, MA, USA). The cell growth inhibition rate was calculated utilizing the following formula: inhibition rate (%) = [Control OD − (Sample OD/Control OD)] × 100, where Control OD is absorbance of negative control and Sample OD is absorbance of the test sample. The IC_50_ values were determined with GraphPad Prism 5 through constructed dose-response curves.

### 2.4. DNA

All DNA constructs were synthesized at GenScript. The cDNA coding for HA-tagged FZD4wt presents an HA epitope allocated downstream of the endoplasmic reticulum targeting sequence of human FZD4 (24). GFP-tagged DVL-1 presents the cDNA coding for GFP in frame with the N-terminus of human DVL-1. In the reporter construct WRE-GFP-wt (24), eight repeats of the optimized TCF/LEF binding sequence (5′-AGATCAAAGGGG-3′) (interspersed with the 5′-GTA-3′ triplet) were produced as already described (REFM). cDNAs were all cloned in expression vector pCDNA3.1.

### 2.5. Inhibition of DVL1 Recruitment by FZD4

HEK293 cells were seeded in 96-well tissue culture plates at a density of 5 × 10 ^3^/well. After 16–24 h seeding, cells were co-transfected with HA-FZD4-wt (0.04 µg/well) and GFP-tagged DVL-1 (0.04 µg/well) using PEI (0.25 µL/well) in a final volume of 100 µL/well. After 24 h of transfection, the medium was replaced and supplemented with the indicated compounds diluted in culture medium. After 24 h of treatment, cells were fixed in 3.7% formaldehyde freshly diluted in PBS for 30 min at RT, quenched with 0.1 M glycine/PBS for 30 min at RT, and washed with PBS. Samples were observed under a Leica confocal fluorescent microscope. Inhibition of DVL recruitment was measured by counting the percentage of cells losing GFP-DVL-1 localization on the PM. Experiments were performed in triplicate, and the standard deviations are indicated.

### 2.6. Inhibition of TCF/LEF Activity

To further prove the inhibitory activity on the WNT-β catenin pathway, we measured the activity of the WNT responsive elements as already described [24]. Briefly, Hek293 cells were seeded on black 96-well Optiplates (PerkinElmer, Waltham, MA, USA) at a density of 5 × 10^3^/well. After seeding for 16–24 h, cells were co-transfected with the cDNA coding for HA-FZD4 (0.04 µg/well) as well as with the reporter construct WRE-GFP (0.04 µg/well) using Lipofectamine. Hek293 cells were plated and treated with different dilutions of compounds **2**, (*S*)-**2** or (*R*)-**1** for 48 h before being fixed and analyzed under a fluorescence microscope. Inhibition of WRE activity was measured by counting the percentage of GFP positive cells. Experiments were performed in triplicate and the standard deviations are indicated.

### 2.7. Statistical Analysis

EC_50_ values were calculated from dose-response data using nonlinear regression analysis in Prism version 6.0 (GraphPad, GraphPad Software, San Diego, CA, USA). All data were analyzed using the two-tailed Student’s *t*-test. A two-way analysis of variance test was used to compare columns. *p* values of <0.05 were considered statistically significant.

### 2.8. Competition Binding Experiments

Equilibrium binding experiments were performed on a Fluoromax-4 single photon counting spectrofluorometer (Jobin-Yvon, NJ, USA). The buffer used was sodium phosphate 50 mM, DMSO 20% *v*/*v*, pH 7.2, and the temperature was set at 25 °C. A constant concentration (1 µM) of a peptide mimicking the C-terminal portion of TMEM88, ranging from residue 153 to residue 159 (TSGKVWV), modified with a dansyl group covalently linked to its N-terminus, was mixed with increasing concentrations of DVL1 PDZ domain (ranging from 2 µM to 38 µM). FRET emissions between 450 and 540 nm at different concentrations of DVL1 PDZ domain were collected as averages of three independent measurements, by exciting the sample at 280 nm in a quartz cuvette with a path length of 1 cm. Experiments were performed in the absence and in the presence of 1 µM and 5 µM constant concentration of (*S*)-**2** or (*R*)-**1** enantiomers. Normalized fluorescence collected at 540 nm was fitted using a hyperbolic function.

### 2.9. Cell Culture and ROS Quantification

HCT116 cells were grown in RPMI-1640 medium (Corning) containing 2 mM L-glutamine and supplemented with 100 IU/mL penicillin/streptomycin and 10% fetal bovine serum (FBS; Sigma-Aldrich, St Louis, MO, USA). HCT116 cells were treated in 96-well-black plates for 48 h with increased concentrations of (*R*)-2 or (*S*)-*2*. Quantification of ROS production was measured using 20 μM of the fluorescent probe 2′,7′-dichlorodihydrofluorescein diacetate DCFDA (Sigma Aldrich). The relative fluorescence emission was followed at 520 nm using VICTOR plate reader (VICTOR™ Multilabel Counter, Perkin Elmer) and was normalized to total protein content using BCA test. Total ROS production was presented as percentage with respect to the control (DMSO treated cells) that was set as 100%. Statistical analysis was performed using one-way ANOVA followed by the Bonferroni post hoc comparison test. *p* < 0.05 was considered significant. Data are representative of three independent experiments with similar results.

## 3. Results

### 3.1. Virtual Screening and Pharmacophore Model

The crucial role played by PDZ domains in the spreading of cellular signals and their aberrant activations in many human diseases makes the PDZ domains very attractive targets for drug discovery [37]. The identification of small molecules able to impair the protein–protein interactions in which PDZs are involved proved to be very profitable, rather than targeting entire signaling cascades, as typically achieved by receptor antagonists [38]. To identify new DVL1 inhibitors, we followed a virtual screening approach by fixing as the main filter a pharmacophoric model. We speculated that compound **2**, the recently reported inhibitor of NHERF1 PDZ1 domain [26], may be able also to bind the DVL PDZ domain. The activity of compound **2** was assayed in HEK293 cells transiently expressing both HA-tagged-FZD4 (HA-FZD4) and the chimeric construct DVL-GFP. In the absence of HA-FZD4, DVL-GFP is localized in punctate intracellular structures known as signalosomes [25]. In the presence of FZD-4 expression, DVL-GFP is recruited on the PM of the cell showing a drastic change in localization. As previously reported [24], drugs acting as inhibitors of FZD4/DVL binding reduce the amount of DVL-GFP recruited on the PM. In dose-response experiments, compound **2** inhibited DVL-GFP binding to HA-FZD4 with an EC_50_ of 5.10 ± 0.09 μM. Thus, we used the information gained from the binding mode of compound **2** to design a pharmacophore model that would lead to the identification of new DVL1 PDZ binders. At the same time, to better distinguish between the inhibitory activities of DVL1 and NHERF1, we added a new query to the pharmacophore model. Actually, comparing the PDZ structures of the studied proteins, we identified a significant difference. The upper part of the carboxylic binding loop was delimited by Tyr24, Ile79 and Val86 for NHERF1, while we observed Leu12, Val75 and Ile81, respectively, for DVL1. These smaller residues of the DVL1 PDZ resulted in a space that might be able to accommodate a hydrophobic group. This peculiar feature of the DVL1 PDZ was included in our pharmacophore model by adding a hydrophobic query to obtain the required selectivity (Appendix A). Thus, the resulting pharmacophore model had 7 queries: 3 aromatics, 2 hydrophobics, 1 H-bond donor and 1 H-bond acceptor. The in-house compound library of near 7000 molecules was fixed as a training set. All compounds in the training set were docked at the DVL1 PDZ binding site. Actually, the docking studies were carried out to achieve a sort of tuned conformational search. Therefore, we managed only the conformations that might bind the studied binding pocket. Thus, for each molecule, we obtained 10 conformations, and all of the proposed binding poses (7000 molecules per 10 conformations) were scored by the fitting to the pharmacophore model. The hydrophobic query that might drive selectivity between NHERF1 and DVL1 was fixed as a must have. The 500 derivatives with the best fitting to the pharmacophore model were visually inspected, and the most promising (10 molecules) were moved to the biological evaluation. Among the tested compounds, the racemic derivative **1** showed the best EC_50_ value of 0.74 ± 0.08 μM for inhibition of the DVL binding to FZD4.

### 3.2. HPLC Separation of the Enantiomers

The enantioseparation of racemate **1** herein described accomplished on the Chiralpak AD chiral stationary phase (CSP) improved substantially the enantioselective HPLC protocol based on the Chirapak IA CSP previously reported [39]. The racemate **1** was resolved into the two constituent enantiomers by semipreparative HPLC on the Chiralpak AD CSP using the mixture *n*-hexane-ethanol-diethanolamine (DEA) 10–90-0.1 (*v*/*v*/*v*) as a mobile phase. The temperature and flow rate were set at 40 °C and 3.0 mL/min, respectively. To study the performance of the semipreparative 250 mm × 10 mm Chiralpak AD column in terms of sample loading, a feed solution was prepared by dissolving 50 mg of racemic sample in an 18 mL ethanol–DEA 100:0.1 (*v*/*v*) mixture. The injection volume of the feed solution was gradually increased until the maximum limit of the sample injection of 5.5 mg. As shown in Appendix A, under these conditions, the enantiomeric peaks, as well as several unknown impurities present in the racemic sample, were nicely separated. Both collected fractions were enantiomerically pure and the quantitative yields were about 85%. By using the immobilized-type Chiralpak IA CSP in combination with the mobile phase ethanol-DEA 100/0.1 (*v*/*v*), the amount of racemate loaded in a single run was about 1 mg. Therefore, through accurately targeted changes in the chromatographic parameters that governed the enantioseparation process, such as the nature of the CSP and composition of the mobile phase, it was possible to optimize the chromatographic recognition and make available, in high enantiopurity state, both antipodes of compound **1**. In accordance with the circular dichroism (CD) properties of the collected enantiomers, the absolute (*R*)-configuration was assigned to the first eluted enantiomer and (*S*)-configuration to the more retained enantiomer [39].

### 3.3. Binding Mode Analysis and Molecular Dynamics

Analyses of the docking proposed binding poses of (*S*)-**1** allowed us to identify key interactions: (i) the dimethyl phenyl moiety was involved in π-cation interaction with Arg69 and Arg72; (ii) the indole ring formed hydrophobic interactions with Val68; (iii) the pyridine ring formed hydrophobic contacts with Ile14 and Val75; (iv) two H-bonds were also observed involving the indole and carboxamide NHs with the Ile16 and Ile14 backbones, respectively. The (*R*)-**1** enantiomer showed a similar binding mode, but with different H-bond patterns, and also the pyridine ring lay in a different sub-pocket (Figure 2).

Molecular dynamics simulations highlighted two additional H-bonds, both with the Ile16 backbone, involving the nitrogen atom of the first carboxamide group and the oxygen atom of the second carboxamide. Concerning the chiral center, we observed that the methyl group of the (*S*)-enantiomer pointed toward the binding site and formed contacts with Leu71 and Arg72, whereas, in the case of the (*R*)-enantiomer, the methyl group pointed toward the solvent exposed area. There was also another relevant difference between the two enantiomers: the pyridine nitrogen atom of the (*S*)-enantiomer formed a contact with the Leu12 backbone through a water molecule bridge, whereas this kind of interaction was not observed for the (*R*)-enantiomer (Appendix A).

### 3.4. PDZ Binding Inhibition

To validate the in silico predictions, we performed equilibrium binding experiments between the DVL1 PDZ domain and a peptide mimicking the C-terminal portion of TMEM88 protein (TSGKVWV) dansylated at its N-terminus. At the concentration of 5 µM, both the (*S*)- and (*R*)-enantiomers abolished the binding between PDZ and TMEM88. At 1 µM concentration, only (*S*)-**1** appeared to increase the Kd of the complex by a factor of two (from 11.5 ± 0.5 to 20.8 ± 0.5 μM), while the (*R*)-enantiomer showed no inhibition of the binding reaction (Kd of 12.3 ± 0.6 μM) (Figure 3). When tested as inhibitors of the NHERF1 PDZ1 domain, compounds (*S*)-**1** or (*R*)-**1** did not show inhibitory activity (data not shown). These experiments demonstrated that (*S*)-**1** and (*R*)-**1** are selective inhibitors of the DVL1 PDZ domain.

### 3.5. In Vitro Activity

The activity of racemic compound **1** and enantiomers (*S*)-**1** and (*R*)-**1** was assayed in HEK293 cells following DVL-GFP recruitment by FZD4. The racemic mixture of compound **1** reduced DVL-GFP recruitment at the PM with an EC_50_ of 0.74 ± 0.08 μM. Dose-response experiments revealed for (*S*)-**1** and (*R*)-**1** an EC_50_ of 0.49 ± 0.11 μM and 29.5 ± 0.9 μM, respectively (Figure 4 and Table 1).

The activity of the compounds was further assayed in HEK293 cells transiently expressing both HA-FZD4 and the WNT reporter construct WRE-GFP (25). The latter presents the coding sequence of GFP under the control of an optimized WNT responsive element (WRE-wt). When WRE-GFP is transiently transfected in cells, it promotes GFP transcription and translation. The amount of GFP produced by each cell (and thus overall cell-fluorescence) depends on the intracellular levels of WNT pathway activity (Figure 5). The racemic mixture **1** reduced overall GFP fluorescence, confirming its inhibitory activity on the WNT/β-catenin pathway with an EC_50_ of 3.46 ± 0.07 μM. Dose-response experiments revealed for (*S*)-**1** and (*R*)-**1** an EC_50_ of 0.49 ± 0.11 μM and 19.49 ± 0.06, respectively (Table 1).

To further prove the WNT-inhibitory activity, we tested **1**, (*S*)-**1** and (*R*)-**1** on a panel of colon cancer cell lines known to depend on the activity of the WNT pathway for their survival. In vitro cultures of colon cancer cells often present mutations either in the *APC* gene or in the β-catenin gene. In the presence of these mutations, the stability of β-catenin is independent of DVL and FZDs, whose pharmacological modulation does not exert any effect on downstream β-catenin activity. Two of these cell lines, colon cancer SW480 and SW620 cells, were used in dose-response experiments. These revealed for all compounds **1** and (*S*)-**1** and (*R*)-**1** moderate EC_50_ values for cell growth inhibition, all falling in the high micromolar range, confirming that in cells where WNT pathway activity is independent of the DVL state, the three compounds do not exert their effect. We thus tested their growth inhibitory activity on the colon cancer cell line HCT116. In terms of APC and β-catenin, these represent the wt counterparts of SW480 and SW260. HCT116 cells express wt APC and are heterozygous for β-catenin, harboring one wild type and one mutant (Ser45del) allele [40]. WNT pathway activity and the survival of HCT116 cells thus depend on DVL and can be affected by FZD and DVL inhibitors. Interestingly, dose-response experiments on HCT116 cells revealed EC_50_ values of cell growth inhibition for compounds **1**, (*S*)-**1** and (*R*)-**1** of 15.2 ± 1.1, 7.1 ± 0.6 and 28.3 ± 1.2 μM, respectively. These results confirm that the three compounds act on DVL to inhibit the WNT pathway. In vitro activities towards colon cancer cells further support our in vitro results for these compounds and point toward their eligibility as interesting lead compounds (Table 2).

### 3.6. ROS Production

Reactive oxygen species (ROS) play important roles in cell growth and differentiation, metabolism and apoptosis [41]. High levels of ROS are known to cause cell death but are also involved in several pathways including WNT signaling, although the mechanism is still largely unknown [42,43,44]. Correlation between the baseline ROS level, mitochondrial DNA copy number and drug sensitivity was reported [45]. We measured the ability of compounds (*S*)-**1** and (*R*)-**1** to generate ROS in HCT116 cells using the fluorescent probe 2′,7′-dichlorodihydrofluorescein diacetate DCFDA. The ROS production (%) upon 48 h treatment was determined with increasing concentrations of (*S*)-**1** or (*R*)-**1**. The results show that ROS production of (*S*)-**1** was superior to (*R*)-**1** at any tested compound concentration (Figure 6). The greatest ROS production was detected at 10 μM; at higher concentrations, ROS production suddenly decreased. The dramatic reduction of ROS production upon (*S*)-**1** 10 μM treatment for 48 h may be correlated to the death of the HCT116 cells through the activation of the mitochondrial pathway [46].

### 3.7. DVL1 Binding Interaction Features

We recognized two key features of the binding to DVL1: (i) the presence of a 6-membered terminal ring, instead of a 5-membered ring, and (ii) the introduction of a methyl group at position 3 of the side chain (Figure 7). The interactions of both indole nucleus and 3,5-dimethylphenyl group were little affected by the modifications of the side chain. The terminal aromatic ring lay in a hydrophobic pocket surrounded by Leu12, Val75 and Ile81 residues; the 6-member rings filled the binding pocket ensuring, more stable interactions compared to the 5-membered heterocycles, mainly due to steric rather than electrostatic reasons. The methyl group at the C3′ of the side chain was modeled for both (*S*) and (*R*) configurations. We observed that the methyl group of both enantiomers pointed toward the β2 helix; only in the case of the (*S*)-configuration, the methyl formed hydrophobic contacts with Leu71 and Arg72, whereas the methyl of the (*R*)-enantiomer was too far from the closest residues to achieve positive contact. The absence of the pyridine basic nitrogen atom abolished the water-mediated H-bond with Leu12 (see Section 3.3).

## 4. Discussion

Colorectal cancers are common cancers and a leading cause of cancer mortality worldwide [47,48]. As the state of the art, surgical resection potentially provides the most important curative option, together with adjuvant systemic chemotherapy and local radiation. In the era of personalized medicine, the identification of key molecular features or pathways that are specific to certain CRC subtypes may represent potential therapeutic targets, enabling the implementation of tailored therapies and better patient management [49].

One of the hallmarks of colorectal cancer is represented by the activation of the WNT pathway and the increased concentration as well as different cellular localization of β-catenin.

Published data offered preclinical proof of concept that WNT/β-catenin modulators represent a valuable opportunity for new chemotherapeutic agents. Nevertheless, the difficulty in finding these modulators arises from the multitude of interconnected signaling branches and proteins involved in the WNT signaling cascade. Moreover, by virtue of this pathway’s importance in physiological conditions, anticancer drugs acting as specific negative modulators of the pathway should be preferred to pure inhibitors of WNT–β-catenin signaling [50,51].

By far, particular attention has been paid to the NHERF1/PDZ1 domain that governs β-catenin membrane recruitment/displacement through a transient phosphorylation switch. NHERF-1 as well as PDZ1 are able to bind FZDs. The activity of the WNT pathway is strongly influenced by the relative expression levels of the two scaffold proteins. Of note, NHERF-1 overexpression inhibits DVL binding to FZD receptors, while NHERF-1 knock-out favors DVL binding to the FZD. So, DVL and NHERF-1 influence each other, showing competition for the cognate receptor [27].

Looking for small molecules able to modulate the effectors of the WNT pathway, we carried out a virtual screening campaign using as main filter a pharmacophore model. A comparison of the PDZ structures of NHERF1 and DVL1 showed that the DVL1 PDZ had an additional sub-pocket. [52] This feature was added to the pharmacophore model, in terms of a new query, to address the desired selectivity.

The virtual screening approach led to the identification of racemic compound **1**. Preliminary docking studies suggested that the S-enantiomer had a qualitatively better binding mode than the R-enantiomer. Then, the racemate enantioseparation was accomplished, and the pure enantiomers were evaluated in vitro and in vivo. Firstly, we measured the binding capabilities of both enantiomers to the DVL1 PDZ. This result confirmed the binding of compounds (*S*)-**1** and (*R*)-**1** to the DVL1 PDZ. In particular, we observed a two-fold better binding of (*S*)-**1** than (*R*)-**1** at the tested concentration of 1μM. More importantly, the binding experiments showed that the tested compounds did not bind the NHERF1 PDZ domain. These results confirmed the goodness of the in silico approach. Then, the compounds (*S*)-**1** and (*R*)-**1** were tested in a short panel of colorectal cancer cells with wild-type APC (HTC116) and mutated APC (SW480 and SW620). Compound (*S*)-**1** showed the better EC_50_ at one-digit micromolar concentration. Furthermore, the higher sensitivity observed for HTC116 confirmed that the antiproliferative activity was due to DVL1 modulations.

The reported data point toward the eligibility of derivative (*S*)-**1** as an interesting lead compound. Furthermore, to our knowledge, this is the first report of a DVL1 modulator with selectivity toward NHERF1. Finally, further studies aimed at improving the inhibitory concentration and assessing potential synergistic effects with known β-catenin inhibitors or FZD modulators will be evaluated and might represent a new weapon in the armamentarium for the fight against colorectal cancer.

## 5. Conclusions

Structure-based virtual screening studies led to the identification of the racemic compound **1** with selective inhibition of DVL1 binding (EC_50_ of 0.74 ± 0.08 μM) toward NHERF1 and inhibition of the WNT pathway with an EC_50_ of 3.46 ± 0.07 μM. In the molecular dynamic simulations, enantiomer (*S*)-**1** (EC_50_ of 0.49 ± 0.11 μM) showed greater affinity for DVL1 compared to the (*R*)-enantiomer. All compounds, 1, (*S*)-**1** and (*R*)-**1**, inhibited HCT116, SW480 and SW620 colorectal cancer cells at micromolar concentration. Most interestingly, these compounds inhibited the growth of HCT116, a cell line that depends on the activity of the WNT signaling pathway for its survival, expressing wild-type APC, with low micromolar EC_50_ values; compounds 1, (*S*)-**1** and (*R*)-**1** showed EC_50_ values of 15.2 ± 1.1, 7.1 ± 0.6 μM and 28.3 ± 1.2, respectively. A higher level of ROS production was observed in HCT116 cells after treatment with (*S*)-**1** at 20 μM for 48 h. These results suggest that (*S*)-**1** exhibits drug-like characteristics and pave the way for the development of new therapeutic agents against WNT-dependent colon cancer. The design and synthesis of new drug derivatives is in progress in our laboratories.

## Figures and Tables

**Figure 1 cancers-14-01358-f001:**
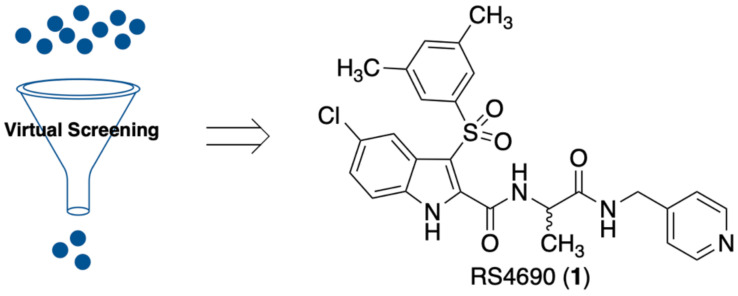
Compound RS4690 (**1**) achieved from virtual screening studies.

**Figure 2 cancers-14-01358-f002:**
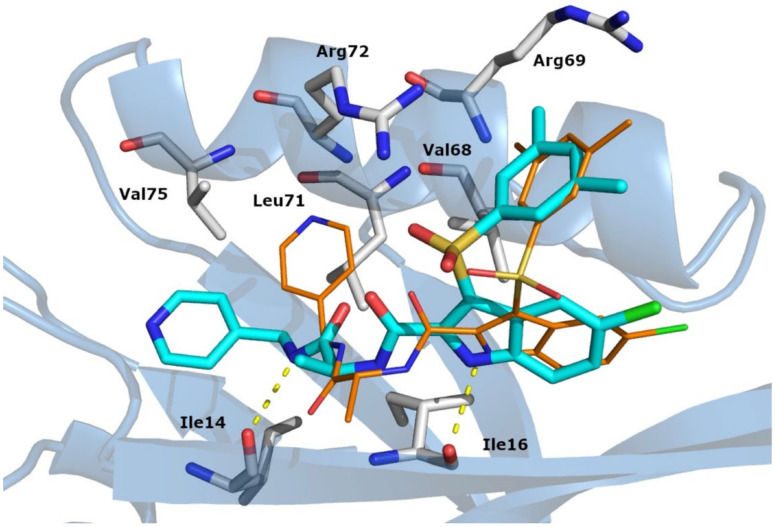
In silico docking results of (*S*)-**1** (cyan) and (*R*)-**1** (orange) in complex with the DVL1 PDZ binding site. Residues involved in interactions are reported as white sticks. The PDZ is depicted as light blue cartoon. H-bonds are reported as yellow dotted lines.

**Figure 3 cancers-14-01358-f003:**
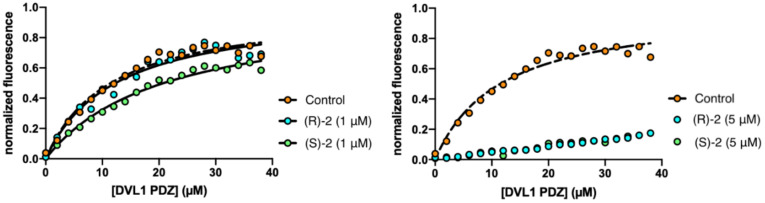
Equilibrium binding experiment between PDZ domain of DVL1 and the C-terminal portion of TMEM88 in the absence and presence of (*S*)-**1** or (*R*)-**1** at 1 µM (**right** panel) and 5 µM (**left** panel). Lines are the best fit for a hyperbolic function.

**Figure 4 cancers-14-01358-f004:**
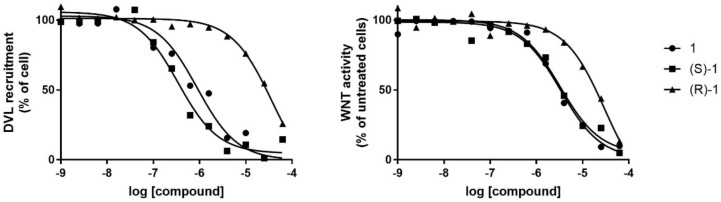
The dose response curves are representative for DVL recruitment inhibition experiment (**left** panel) and WNT pathway activity measurement (**right** panel).

**Figure 5 cancers-14-01358-f005:**
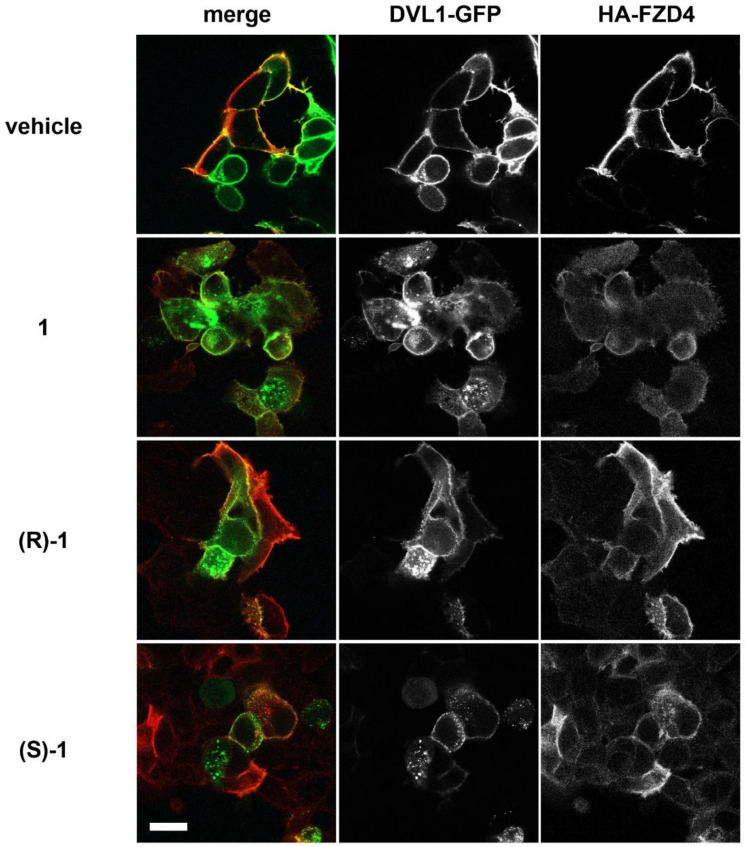
Confocal immunofluorescence showing the recruitment of DVL1-GFP (green) by HA-FZD4 (red) at the PM of HEK293 cells in the presence of 7 μM of compound **1** (racemic mixture), (*S*)-**1**, (*R*)-**1** or the corresponding volume of vehicle (DMSO). (Magnification bar = to 14 μm).

**Figure 6 cancers-14-01358-f006:**
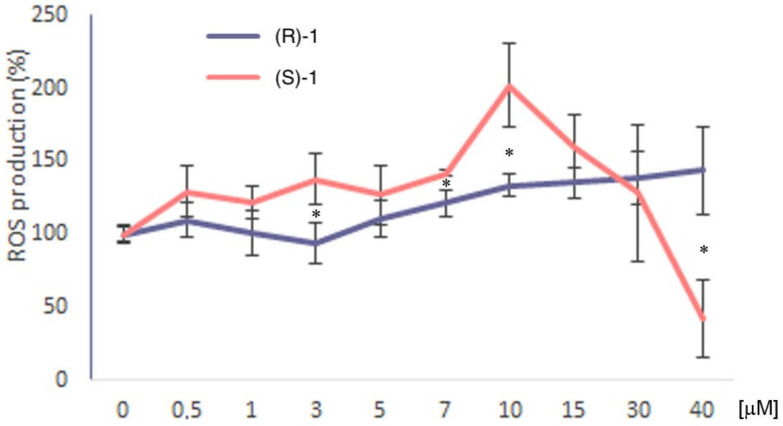
ROS production in HCT116 cells upon treatment with different concentrations of (*R*)-**2** or (*S*)-**2** for 48 h. The asterisks show the differences between the effects induced by the two drugs.

**Figure 7 cancers-14-01358-f007:**
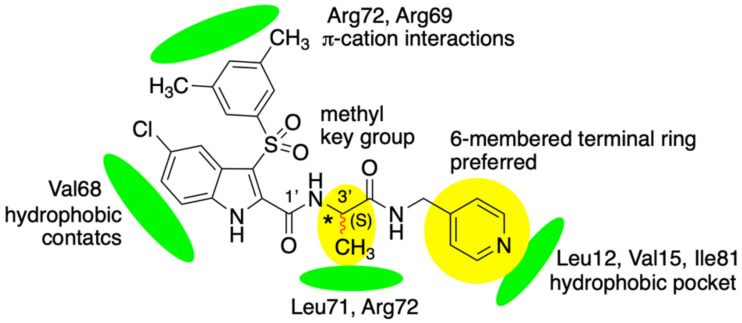
Key contacts of compound (*S*)-**2** in the DVL1 PDZ binding pocket. H-bonds are not shown for the sake of clarity.

**Table 1 cancers-14-01358-t001:** DVL Binding and WNT Pathway Inhibition by Compounds **1**, (*S*)-**1**, (*R*)-**1** and **2**.

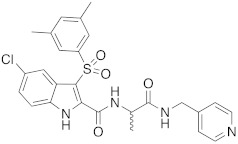		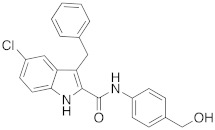
**1**		**2**
	**EC_50_ ± SD (μM) ^a^**	**EC_50_ ± SD (μM) ^b^**
Compound	DVL1 Binding Inhibition	WNT Pathway Inhibition
**1**	0.74 ± 0.08	3.46 ± 0.07
(*S*)-**1**	0.49 ± 0.11	3.09 ± 0.05
(*R*)-**1**	29.5 ± 0.9	19.49 ± 0.06
**2**	5.10 ± 0.09	n.d.

^a^ Inhibition of DVL1 binding. ^b^ Inhibition of the WNT pathway.

**Table 2 cancers-14-01358-t002:** Growth Inhibition of SW680, SW620 and HCT116 Human Colon Carcinoma Cell Lines by Racemate **1** and Enantiomers (*S*)-**1** and (*R*)-**1**.

	EC_50_ (μM)/Cell Lines		
Compound	SW480 ^a^	SW620 ^a^	HCT116 ^b^
**1**	39.17 ± 1.58	38.54 ± 1.6	15.2 ± 1.1
(*S*)-**1**	54.95 ± 2.2	45.9 ± 2.0	7.1 ± 0.6
(*R*)-**1**	59.47 ± 2.1	54.75 ± 1.9	28.3 ± 1.2

^a^ Incubation time was 72 h. ^b^ Incubation time was 48 h.

## Data Availability

The datasets generated and/or analyzed in this study are available from the corresponding author upon reasonable request.

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
