# Peer review of "Anticancer Activity of (S)-5-Chloro-3-((3,5-dimethylphenyl)sulfonyl)-N-(1-oxo-1-((pyridin-4-ylmethyl)amino)propan-2-yl)-1H-indole-2-carboxamide (RS4690), a New Dishevelled 1 Inhibitor"

_cancers, 2022, doi:10.3390/cancers14051358_

Round 1

Reviewer 1 Report

The authors reported anticancer activity RS4690 in SW480, SW620 and HCT116 cell lines. The results are interesting. The authors identified the racemic compound RS4690 from structure based virtual screening, but also failed to report number of compounds screened. Growth inhibition data shows that racemic compound 1 has low EC50 value in comparison to (S)-1 and (R)-1 in SW480 and SW620 cell lines. The authors did not discuss why racemic is has low EC50 than any of the enatiomeric pure compounds. The discussion section should be revised and in depth discussion of the significance of the work would be beneficial to impact of the manuscript.  

Few other things: Please report the structure of  compound in the manuscript, it is easier to the reader to compare. Quality and presentation of docking images could be improved. 

Author Response

The authors reported anticancer activity RS4690 in SW480, SW620 and HCT116 cell lines. The results are interesting.

Reviewer 1.1: The authors identified the racemic compound RS4690 from structure based virtual screening, but also failed to report number of compounds screened.

Reply: The number of the screened molecules together with some details of the virtual screening campaign were added to the manuscript. (Lines 297-305)

Reviewer 1.2: The authors did not discuss why racemic is has low EC50 than any of the enatiomeric pure compounds.

Reply: We thank the reviewer for this question. The derivatives 1, (S)-1 and (R)-1 showed comparable EC50 for SW480 and SW629 cells, with slight better values for the racemic mixture. On the contrary, the best EC50values were measured for HCT116 cells is in accordance with the proposed mechanism of action. It was decided to do not discuss the EC50 difference measured for SW480 and SW620 cells and to consider the three tested compounds equipotent. Indeed, it should be taken into account that the ANOVA test show no significant differences, and it could be argued that this is due to the low efficiency of these compounds on the SW480 and SW620 cell line.

Reviewer 1.3: The discussion section should be revised and in depth discussion of the significance of the work would be beneficial to impact of the manuscript.

Reply: The discussion was completely rephased

Reviewer 1.4: Please report the structure of compound 2 in the manuscript, it is easier to the reader to compare. Quality and presentation of docking images could be improved.

Reply: The structure of the compounds 1 and 2 were added to table 1

Reviewer 2 Report

The manuscript contains interesting results, but the presentation, data organization, description of results and discussions are not good.

Specific comments:

The title is unclear, selectivity toward NHERF1 is not mentioned/explained in the abstract. Do you mean selectivity over NHERF1? In addition, NHEFR1 is less known protein, its use in the title is not recommended.

Consider rephrasing the lines 49-50 (Compound (S)-1 inhibited the growth of HCT116 cells that does not present the APC mutation, ...). Do you mean HCT116 cells expressing wild type APC?

Punctuatin is not used properly (...APC mutation, with EC50 value...).

Screening campaign is not described sufficiently.

Line 368: Figure 4 seems to show different data. Dose-response examples/curves would be much helpful.

Lines 378-381: Consider rephrasing to be clearer. WHy do you mention KRAS and BRAF, which regulate different signaling pathways? There are far more mutations in HCT116 cells than these two. Wwhat is implication of catenin heterozygosity on the WNT/APC pathway?

Figure 3: Are the data points averages from more measurements? Method is not described.

Used cancer cell lines should be clearly characterized in terms of activity/dependence/mutations of WNT signaling pathway members. Why HCT116 cells are more sensitive to the compound than other two used cell lines? I would recommend to assay compounds on more cell lines with defined WNT pathway member mutations and wt counterparts, ideally on isogenic pairs.

Figure 4 caption: Magnification bar is not shown.

Table 2 caption: typo (Hman Colon) 

Analysis of ROS production is meaningless, what is the relation to WNT signaling? The changes observed after 48 h could be  only an indirect consequence of cytotoxicity. More importantly, the ROS values seem to fluctuate too much and randomly and the differences are only at three discrete doses. Conclusions are not really valid.

Line 403: typo (mM)

Chapter 3.8 is nearly meaningless. The in silico predictions have usually low value, especially when shown only for one compound without any experimental validation. I recommend to remove from the main text.

Line 492: grammar error (screening studies led to identify racemic)

First two paragraphs are not really important in the context of the medchem/biochemical paper. Focudes discussion on molecular mechanism of action is preferred.

Line 482: remove capitalization (Plasma Membrane)

Author Response

Reviewer 2

The manuscript contains interesting results, but the presentation, data organization, description of results and discussions are not good.

Specific comments:

Reviewer 2.1: The title is unclear, selectivity toward NHERF1 is not mentioned/explained in the abstract. Do you mean selectivity over NHERF1? In addition, NHEFR1 is less known protein, its use in the title is not recommended.

Reply: The title of the manuscript was modified following the reviewer suggestion

Reviewer 2.2: Consider rephrasing the lines 49-50 (Compound (S)-1 inhibited the growth of HCT116 cells that does not present the APC mutation, ...). Do you mean HCT116 cells expressing wild type APC?

Reply: The sentence was update following the reviewer suggestion

Reviewer 2.3: Punctuatin is not used properly (...APC mutation, with EC50 value...).

Reply: The all manuscript was carefully checked, and typo and punctation were fixed

Reviewer 2.4: Screening campaign is not described sufficiently.

Reply: The description of the virtual screening procedure was updated. (Lines 295-307)

Reviewer 2.5 Line 368: Figure 4 seems to show different data. Dose-response examples/curves would be much helpful.

Reply: Actually, there was a misplacement of reference for figures in the text. We are sorry for that. The figure was fixed in our revised version. Dose response curves were added (Figure 4) to the revised version of our manuscript.

Reviewer 2.6 Lines 378-381: Consider rephrasing to be clearer. WHy do you mention KRAS and BRAF, which regulate different signaling pathways? There are far more mutations in HCT116 cells than these two. What is implication of catenin heterozygosity on the WNT/APC pathway?

Reply: Following the reviewer suggestion the weakness was fixed and mention to KRAS and BRAF removed, also see reply to point 8

Reviewer 2.7 Figure 3: Are the data points averages from more measurements? Method is not described.

Reply: All data points presented in the graph represent the average of three independent measurements. We added this in the Materials and Methods section. (lines 251-252)

Reviewer 2.8 Used cancer cell lines should be clearly characterized in terms of activity/dependence/mutations of WNT signaling pathway members. Why HCT116 cells are more sensitive to the compound than other two used cell lines? I would recommend to assay compounds on more cell lines with defined WNT pathway member mutations and wt counterparts, ideally on isogenic pairs.

Reply: We are sorry to not have been clearer in our explanation. In the revised version of our manuscript, we better explain the rationale underpinning the choice of these cell lines for our tests. All the cancer cell line we have used in our manuscript present a constitutively active WNT pathway and present defined WNT pathway member mutations. In terms of APC and beta-catenin SW480, SW629 and HCT116 represent mutants and wt counterpart. SW480 and SW629 present a mutation in the APC gene. Inhibitors of DVL do not exert any effect on this type cells, because whatever is the state of DVL, their WNT pathway will be active anyway. HCT116 present a wt APC gene and one wt allele of beta catenin. In these cells WNT pathway can be switched off by DVL inhibitors. The absence of toxicity of compounds 1, (S)-1 and (R)-1 on SW480, SW629 and their activity in HCT116 confirmed again that these molecules target DVL.

Reviewer 2.9 Figure 4 caption: Magnification bar is not shown.

Reply: Magnification bar was added to the revised figure

Reviewer 2.10 Table 2 caption: typo (Hman Colon)

Reply: The typo was fixed

Reviewer 2.11a Analysis of ROS production is meaningless, what is the relation to WNT signaling? The changes observed after 48 h could be  only an indirect consequence of cytotoxicity.

Reply: The connection between WNT signaling and ROS production is extensively reported in several models but the mechanism is still largely unknown and seems to involve the import of calcium in the mitochondria (1]Regulation of the Wnt/β-catenin pathway by redox signaling. Hendrik C. Korswagen. Dev Cell. 2006; 10(6):687-8. 2] ROS, Notch, and Wnt Signaling Pathways: Crosstalk between Three Major Regulators of Cardiovascular Biology. C. Caliceti, et al. Biomed Res Int. 2014; 318714; 3] Ca2-mediated Mitochondrial Reactive Oxygen Species Metabolism Augments Wnt/-Catenin Pathway Activation to Facilitate Cell Differentiation. Tareck Rharass,et al. J Biol Chem. 2014; 289(40):27937-51). We have changed the text and added the relative references to better clarify the point. (Lines 446-448)

Reviewer 2.11b More importantly, the ROS values seem to fluctuate too much and randomly and the differences are only at three discrete doses. Conclusions are not really valid.

Reply: The asterisks in Figure 6 show the differences between the effects induced by the two drugs, we have also inserted the asterisks relating to the comparison with respect to the control (no drugs). At concentrations higher than 10 micromolar the cytotoxicity of (S)-1 influences the ROS production; indeed, we observed a quick reduction of ROS production mainly related with cells death. Nevertheless, the data with respect to the control are significant from 7 to 40 micromolar using each of the drugs

Reviewer 2.12 Line 403: typo (mM)

Reply: the mistyping was correct

Reviewer 2.13 Chapter 3.8 is nearly meaningless. The in silico predictions have usually low value, especially when shown only for one compound without any experimental validation. I recommend to remove from the main text.

Reply: The Chapter 3.8 was removed

Reviewer 2.14 Line 492: grammar error (screening studies led to identify racemic)

Reply: the error was fixed

Reviewer 2.15 First two paragraphs are not really important in the context of the medchem/biochemical paper. Focudes discussion on molecular mechanism of action is preferred.

Reply: All paragraphs of the manuscript were checked and updated following the suggestions of the reviewers

Reviewer 2.16 Line 482: remove capitalization (Plasma Membrane)

Reply: capital letters were changed to small letters

Round 2

Reviewer 1 Report

The authors amended manuscript to improve quality of the paper. I recommend this manuscript for publication in current form.